

# Tea intake or consumption and the risk of dementia: a meta-analysis of prospective cohort studies

Ning Jiang[1,*], Jinlong Ma[2,*], Qian Wang[3], Yuzhen Xu[4] and Baojian Wei[1]

[1] School of Nursing, Shandong First Medical University & Shandong Academy of Medical Sciences, Taian, Shandong, China
[2] Yanbian University, Yanbian, Jilin, China
[3] Postdoctoral Workstation, Department of Central Laboratory, The Affiliated Taian City Central Hospital of Qingdao University, Taian, Shandong, China
[4] The Second Affiliated Hospital, Shandong First Medical University, Taian, Shandong, China
* These authors contributed equally to this work.

Corresponding authors
Yuzhen Xu,
xuyuzhen@sdfmu.edu.cn
Baojian Wei, bjwei@sdfmu.edu.cn

## ABSTRACT

**Purpose:** Dementia affects as many as 130 million people, which presents a significant and growing medical burden globally. This meta-analysis aims to assess whether tea intake, tea consumption can reduce the risk of dementia, Alzheimer's disease (AD) and Vascular dementia (VD).

**Patients and methods:** Cochrane Library, PubMed and Embase were searched for cohort studies from inception to November 1, 2022. The Newcastle Ottawa Quality Assessment Scale (NOS) was applied to evaluate the risk of bias of the included studies. We extracted the data as the relative risks (RRs) for the outcome of the interest, and conducted the meta-analysis utilizing the random effect model due to the certain heterogeneity. Sensitivity analysis were performed by moving one study at a time, Subgroup-analysis was carried out according to different ages and dementia types. And the funnel plots based on Egger's and Begger's regression tests were used to evaluate publication bias. All statistical analyses were performed using Stata statistical software version 14.0 and R studio version 4.2.0.

**Results:** Seven prospective cohort studies covering 410,951 individuals, which were published from 2009 and 2022 were included in this meta-analysis. The methodological quality of these studies was relatively with five out of seven being of high quality and the remaining being of moderate. The pooling analysis shows that the relationship between tea intake or consumption is associated with a reduced risk of all-cause dementia (RR = 0.71, 95% CI [0.57–0.88], $I^2$ = 79.0%, $p < 0.01$). Further, the subgroup-analysis revealed that tea intake or consumption is associated with a reduced risk of AD (RR = 0.88, 95% CI [0.79–0.99], $I^2$ = 52.6%, $p = 0.024$) and VD (RR = 0.75, 95% CI [0.66–0.85], I = 0.00%, $p < 0.001$). Lastly, tea intake or consumption could reduce the risk of all-cause dementia to a greater degree among populations with less physical activity, older age, APOE carriers, and smokers.

**Conclusion:** Our meta-analysis demonstrated that tea (green tea or black tea) intake or consumption is associated with a significant reduction in the risk of dementia, AD or VD. These findings provide evidence that tea intake or consumption should be recognized as an independent protective factor against the onset of dementia, AD or VD.

# INTRODUCTION

Dementia is a neurodegenerative disease with cognitive deficits, which has become an increasing public health concern and poses heavy burden on global health systems (*Wang, Song & Niu, 2022*). *World Health Organization (2021)* forecasts that dementia cases might rise from 55.2 million in 2022 to 130 million in 2050 globally. Based on the country-specific value of statistical life years, the global economic burden of Alzheimer's disease and related dementias (ADRDs) was an estimated $2.8 trillion, which would skyrocket to a whopping $16.9 trillion in 2050, constituting 65% of economic burden in low- and middle-income countries (*Nandi et al., 2022*). The two major subtypes of dementia are Alzheimer's disease (AD) and vascular dementia (VD), with AD accounting for nearly two-thirds and VD accounting for approximately 25% (*Ferri et al., 2005*; *Corriveau et al., 2017*). Despite extensive research on dementia in the past decades, the current therapeutic methods of dementia still have their own limitations, which including pharmacologic and nonpharmacologic approaches. There is no completely alternate pharmacologic methods and existing drugs have their respective side effects, with poor compliance (*Knight et al., 2018*; *Parsons et al., 2021*). Nonpharmacologic approaches, including cognitive therapy, music therapy, horticultural therapy and physical exercise, require a significant amount of time, effort and money (*The Lancet Neurology, 2022*). Thus, prevention of dementia has become increasingly crucial.

Several potential preventable risk factors of dementia have been identified by the 2020 *Lancet Commission*, among which dietary factors play important roles (*Livingston et al., 2020*). Beverage is the most acceptable dietary modification, because it doesn't affect other dietary habits. Tea is a beloved beverage enjoyed in many cultures worldwide, which consumes in staggering quantities of more than two billion cups daily (*Kochman et al., 2020*). Experimental evidences indicate that caffeine and tea polyphenols contained in tea have neuroprotective effects, such as anti-inflammatory, or anti-oxidant effects (*Ma et al., 2016*; *Panza et al., 2015*; *Feng et al., 2018*). In addition, research reveals that tea may protect against AD, through reducing amyloid-β (Aβ) in the brain (*Arendash & Cao, 2010*; *Polito et al., 2018*). A prospective study shows that green tea is associated with reduced risk of cognitive decline (*Noguchi-Shinohara et al., 2014*). Another cohort research indicates that regular tea consumption, especially black and oolong teas, relate to lower risks of cognitive impairment and decline (*Ng et al., 2008*). Accordingly, tea consumption has correlation with dementia.

Despite several studies have shown that tea protects against dementia, the sample sizes of these studies are small and of great limitations. So there still need a large-scale and rigorous prospective cohort to clarify the association between tea and dementia. Therefore, we conducted this meta-analysis of published prospective studies which contained 410,951 participants to further examine the relationship between tea consumption and dementia.

## METHODS

We registered a standard protocol, developed before study selection, for all steps of this meta-analysis on PROSPERO platform, and the approval number for registration is CRD42022369707. In addition, we followed the Meta-analysis of Observational Studies in Epidemiology (MOOSE) reporting guideline to present this meta-analysis.

### Data sources

Four international databases, Cochrane Library, PubMed, Web of Science and Embase were searched without any restrictions from inception to November 1, 2022.
To comprehensively and accurately gather relevant literature, we employed specific keywords for retrieval. The search terms and their variations utilized in the search included "dementia", "Alzheimer's disease", "vascular dementia," and "tea". For a detailed search strategy, please refer to Tables S1–S4. Additionally, we thoroughly examined the reference lists of retrieved studies and consulted previous high-quality reviews to identify any additional eligible studies.

### Eligibility criteria

Eligible criteria: (a) Events: dementia, AD, or VD; (b) Exposure: the intake or consumption of tea, including tea drinking and the type of tea. The dosages of tea are not limited; (c) Comparison: Non tea drinking, intake or consumption population; (d) Outcomes: hazard ratios (HRs), relative risks (RRs), odds ratios (ORs) or variance of the estimates of the risk of dementia, AD or VD, and description of adjustment for potential confounders; (e) Study design: prospective cohort study.

Exclusion criteria: (a) meeting abstracts, letter to editorials; (b) duplicate publication; (c) incomplete data; (d) no interested outcomes.

### Study selection

We import the initial records retrieved from these databases into NoteExpress reference management software, and use the software's own duplicate checking function in combination with manual screening to eliminate duplication. Thereafter, two reviewers (Ma Jinlong and Gao Shuang) read the title and abstract of the initial records respectively, and excluded irrelevant records. Read the rest of the full text, and determine the final literature for meta-analysis according to the pre-established inclusion and exclusion criteria. Any disputes were resolved through group discussion.

### Data extraction

We designed a data extraction form in Excel software (Microsoft Corporation, Redmond, WA, USA). Two authors (Ning Jiang and Jinlong Ma) independently extracted information from eligible cohorts. The following data was obtained from each study: name of first author, publication year, sample size, country, numbers of dementias, follow up time, age, outcomes, confounders, exposure and type of tea. The extracted data is cross-checked, and disagreements were resolved through discussed with the third reviewer (Wei Baojian).

## Study quality

The Newcastle Ottawa Quality Assessment Scale (NOS) was used to assess the methodological quality of the included cohort studies (Available at: http://www.ohri.ca/programs/clinical_epidemiology/oxford.asp). NOS assessed the quality of the cohort studies from three dimensions: selection, comparison, and results. The score ranges from 0 to 9, and the higher the score, the higher the quality. NOS score equals or higher than 7, 4 ~ 6 and 0 ~ 3 represent high, medium and low quality respectively.

## Data synthesis

According to previous high-quality observational meta-analysis, (*Zhao et al., 2022*) if a study provided HR instead of RR, we used the formula (RR = $(1 - e^{HR * \ln(1 - r)})/r$; r: the dementia rate) to covert HR to RR; For studies that reported odds ratios (OR) instead of RR, we interpreted the OR as RR when the risk of fracture was below 20%. In cases where the risk exceeded 20%, we used the generic inverse variance method to calculate the pooled RR. Heterogeneity was assessed using the chi-square test and $I^2$ value, with significance set at $p < 0.1$ or $I^2 > 50\%$, indicating substantial heterogeneity and prompting the adoption of a random-effects model. Conversely, a fixed-effect model was employed when heterogeneity was not significant (*Higgins et al., 2003*). Sensitivity analysis was conducted to verify the robustness of the meta-analyses and explore potential sources of heterogeneity. Subgroup analysis was performed based on different age groups and dementia types. However, due to variations in the measurement standards of tea intake, cup sizes, doses, and mixed tea types across the included studies, subgroup analysis according to different tea types and doses could not be carried out. Finally, to detect publication bias, funnel plots, and Egger's and Begg's regression tests were employed (*Egger et al., 1997*). The Stata software (version 14) and R studio (version 4.2.0) was used to conduct the data analysis.

# RESULT

## Literature search

A comprehensive search of four major medical databases yielded a total of 1,545 records retrieved. After removing 343 duplicate records, 1,177 records were excluded based on their title and abstract. We obtained the full text of remaining 25 records for closer inspection, and seven studies were included in this meta-analysis (*Hu et al., 2022*; *Matsushita et al., 2021*; *Chuang et al., 2019*; *Fischer et al., 2018*; *Tomata et al., 2016*; *Noguchi-Shinohara et al., 2014*; *Eskelinen et al., 2009*). Details of literature screening are presented in PRISMA flow chart (Fig. 1).

## Study characteristics and methodology quality

Seven prospective cohort studies were included, published from 2009 to 2022, with a total of 410,951 participants. The follow-up period in these studies lasted between 4.9 and 21 years, with an average of 9 years (detailed characteristics are presented in Table 1). Out of the participants, 7,382 individuals were diagnosed with dementia during or by the end of the study period. All studies looked at the relationship between tea consumption and dementia risk, with four of them examining the risk of AD and two examining the risk

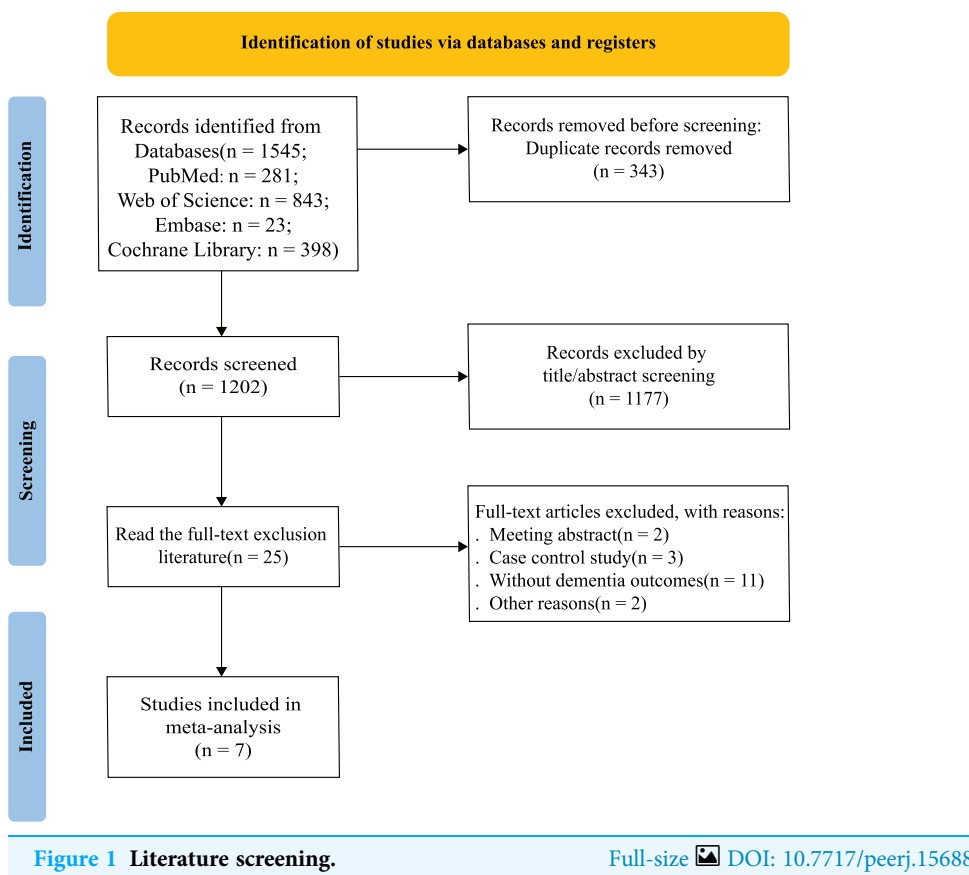

**Figure 1  Literature screening.**

of VD. The International Classification of Diseases (ICD) was used in most studies to diagnose dementia. The data analysis of all seven studies was rigorous, with adjustments made for factors such as age, gender, education, disease history, and other variables that could affect the results.

The methodological quality of these cohort studies was evaluated using the NOS scale. Out of the seven studies, two were considered to be of moderate quality. And, the other five studies scored above seven points on the scale, indicating that they were of relatively high quality. Overall, the studies were conducted with sound methodology and produced reliable results.

## Meta-analysis of all-cause dementia

Seven studies (*Hu et al., 2022*; *Matsushita et al., 2021*; *Chuang et al., 2019*; *Fischer et al., 2018*; *Tomata et al., 2016*; *Noguchi-Shinohara et al., 2014*; *Eskelinen et al., 2009*) have reported the relationship between tea intake or consumption and the risk of all-cause dementia (RR = 0.71, 95% CI [0.57–0.88], $I^2$ = 79.0%, $p < 0.01$). This suggested that tea intake or consumption can reduce the risk of all-cause dementia by 29% (Fig. 2). As there was some heterogeneity, the random effect model was used for the meta-analysis, and sensitivity analysis was performed by removing each study one by one. No single study led to a reversal of the overall result (Fig. 3). Overall, the findings of the analysis support the idea that tea intake may be helpful in reducing the risk of all-cause dementia.

**Table 1  Characteristics of studies included in the review.**

| Author | Year | Data source | Sample size | No. of dementia | Follow up time | Baseline age (years) | Diagnosis of dementia | Outcomes | Confounders adjusted | Tea type | Main findings | NOS score |
|---|---|---|---|---|---|---|---|---|---|---|---|---|
| Hu et al. (2022) | 2022 | UK Biobank | 377,592 | 5,122 | 9 years | 58.49 ± 6.83 | ICD-9 and ICD-10 | All caused dementia, AD, VD | Age, sex, ethnicity, TDI, education, BMI, typical sleep duration, smoking status, alcohol status, total consumption of vegetables, total consumption of fruit, total consumption of fish and APOE4 status | Black tea Green tea | A U-shaped association between tea consumption and dementia risk, and the consumption of around three cups per day showed the strongest protective effect | 9 |
| Schaefer et al. (2022) | 2022 | UK Biobank | 351,436 | 4,270 | 12 years | 38 ~ 73 | ICD-10 | All caused dementia | Age, body fat | Black tea Green tea | Moderate-to-high tea intake was negatively associated with incident dementia | 6 |
| Zhang et al. (2021) | 2021 | UK Biobank | 365,682 | 5,079 | 11.4 years | 50 ~ 74 | ICD-10 | All caused dementia, AD, VD | Sex, age, ethnicity, qualification, income, BMI, physical activity, alcohol status, smoking status, diet pattern, consumption of sugar-sweetened beverages, HDL, LDL, history of cancer/diabetes/CAD/hypertension | Black tea Green tea | Intake of coffee alone or in combination with tea was associated with lower risk of poststroke dementia | 8 |
| Matsushita et al. (2021) | 2021 | Japan | 13,757 | 309 | 8 years | 40 ~ 74 | / | All caused dementia | BMI, physical activity, energy, smoking, drinking, and disease history | Green tea | The association between green tea consumption and reduced dementia risk was significant only in the 60–69 years age subgroup | 6 |
| Chuang et al. (2019) | 2019 | China | 1,436 | 260 | 11.04 years | ≥65 | ICD-9 | All caused dementia | Age, sex, education, baseline cognition, body mass index, stroke history, diastolic blood pressure, inflammation status, and stroke occurrence | Tea | Higher intakes of both tea and fish were associated with an even lower risk od dementia | 7 |

Jiang et al. (2023), *PeerJ*, DOI 10.7717/peerj.15688

| Table 1 (continued) | | | | | | | | | | | |
|---|---|---|---|---|---|---|---|---|---|---|---|
| **Author** | **Year** | **Data source** | **Sample size** | **No. of dementia** | **Follow up time** | **Baseline age (years)** | **Diagnosis of dementia** | **Outcomes** | **Confounders adjusted** | **Tea type** | **Main findings** | **NOS score** |
| *Fischer et al. (2018)* | 2018 | Germany | 2,622 | 418 | 10 years | ≥75 | ICD-10 | All caused dementia, AD | Age, gender, BMI, education, APOE'4 carrier status, smoking status, physical activity score, depression, hypercholesterolemia, | Black tea | Only higher red wine intake not tea was associated with a lower incidence of AD | 7 |
| *Tomata et al. (2016)* | 2016 | Japan | 13,645 | 1,186 | 5.7 years | ≥65 | Dementia Scale | All caused dementia | Age, sex, education level, smoking, alcohol drinking, BMI, history of disease | Green tea | Green tea consumption is significantly associated with a lower risk of incident dementia | 7 |
| *Noguchi-Shinohara et al. (2014)* | 2014 | Japan | 723 | 26 | 4.9 years | >60 | DSM-III-R | All caused dementia | Age, sex, history of hypertension, diabetes mellitus, hyperlipidemia, education, and ApoE4 carrier status | Green tea | No association between coffee or black tea consumption and the incidence of dementia or MCI | 6 |
| *Eskelinen et al. (2009)* | 2009 | Finland | 1,409 | 61 | 21 years | 65 ~ 79 | DSM-IV | All caused dementia, AD | Age, sex, education, follow-up time and community of residence, | Tea | Tea drinking was relatively uncommon and was not associated with dementia/AD | 7 |

**Note:**
ICD, International Classification of Diseases; BMI, body mass index; HDL, high-density lipoprotein; LDL, low density lipoprotein; CVD, cardiovascular arterial disease; DSM-III-R/DSM-IV, Diagnostic and Statistical Manual of Mental Disorders, Third Edition, Revised.

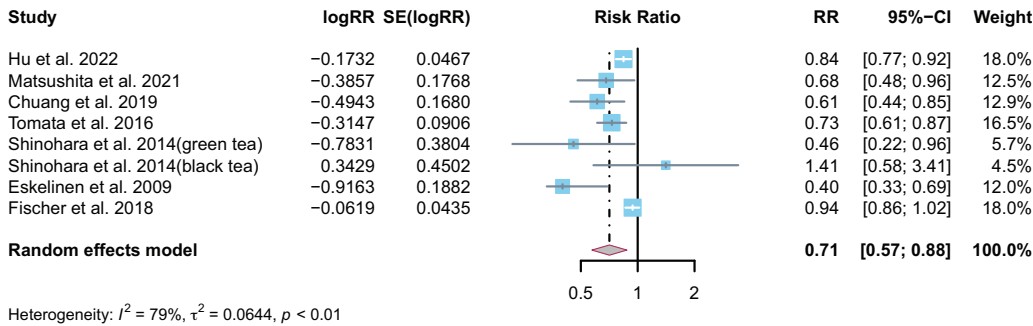

**Figure 2** Forest map of tea intake or consumption and risk of all-cause dementia.

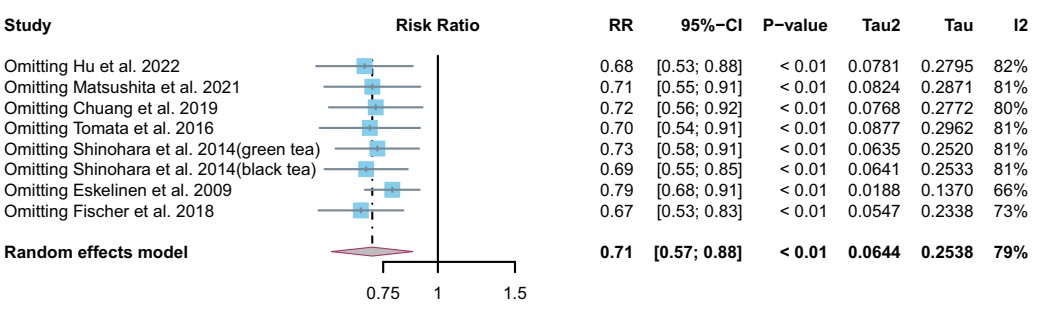

**Figure 3** Results of sensitivity analysis.

## Subgroup analysis

Four studies (*Hu et al., 2022*; *Fischer et al., 2018*; *Eskelinen et al., 2009*; *Zhang et al., 2021*) have investigated the link between tea intake or consumption and the risk of Alzheimer's disease (AD) (Table 2, RR = 0.88, 95% CI [0.79–0.98], $I^2$ = 52.6%, $p$ = 0.024), indicating that tea intake or consumption can reduce the risk of AD by 12%. When it comes to vascular dementia (VD), Two cohorts (*Hu et al., 2022*; *Zhang et al., 2021*) have explored the relationship between tea intake or consumption and the risk of VD (Table 2, RR = 0.75, 95% CI [0.66–0.85], $I^2$ = 0.00%, $p$ < 0.001), suggesting that tea intake or consumption has been associated with a greater reduction in the risk of VD compared to AD.

Six studies (*Chuang et al., 2019*; *Fischer et al., 2018*; *Tomata et al., 2016*; *Noguchi-Shinohara et al., 2014*; *Eskelinen et al., 2009*) have investigated that tea intake or consumption in people aged 65 or older, and the result showed that tea intake or consumption could reduce the risk of all-cause dementia by 32% (Table 2, RR = 0.68, 95% CI [0.50–0.92], $I^2$ = 84%, $p$ < 0.01). And two studies (*Hu et al., 2022*; *Matsushita et al., 2021*) reported the result for those younger than 65 years old (Table 2, RR = 0.81, 95% CI [0.69–0.95], $I^2$ = 26%, $p$ = 0.25). This finding support that the benefits of tea intake or consumption for reducing the risk of dementia are more prominent among older adults.

Three studies (*Chuang et al., 2019*; *Tomata et al., 2016*; *Noguchi-Shinohara et al., 2014*) included over 40% APOE ε4 carriers. In these studies, tea intake can reduce the risk of dementia by remarkable 30% (Table 2, RR = 0.70, 95% CI [0.43–1.15], $I^2$ = 90%, $p$ < 0.01). In two studies (*Hu et al., 2022*; *Eskelinen et al., 2009*) that included less than 40% APOE ε 4

**Table 2 Results of subgroup.**

| Subgroups | Included studies | RR (95% CI) | Heterogeneity | |
|---|---|---|---|---|
| | | | $I^2$ (%) | p-value |
| *Dementia type* | | | | |
| AD | 4 | 0.88 [0.79–0.98] | 52.6 | 0.024 |
| VD | 2 | 0.75 [0.66–0.85] | 0.0 | <0.001 |
| *Age* | | | | |
| ≥65 | 5 | 0.67 [0.51–0.87] | 75.9 | 0.003 |
| >65 | 1 | 0.78 [0.69–0.88] | / | <0.001 |

carriers, tea intake can reduce the risk of dementia by 22% (Table 2, RR = 0.78, 95% CI [0.26–2.35], $I^2$ = 73%, $p$ = 0.06). The results suggest that tea consumption may offer more protection in those with a high genetic risk.

Three studies (*Hu et al., 2022*; *Matsushita et al., 2021*; *Tomata et al., 2016*) included over 80% tea drinkers who drink tea every day. In these studies, tea intake can reduce the risk of dementia by 21% (Table 2, RR = 0.79, 95% CI [0.69–0.89], $I^2$ = 32%, $p$ = 0.23). In four studies (*Chuang et al., 2019*; *Fischer et al., 2018*; *Noguchi-Shinohara et al., 2014*; *Eskelinen et al., 2009*) that included less than 80% tea drinkers who drink tea every day, tea intake can reduce the risk of dementia by 40% (Table 2, RR = 0.60, 95% CI [0.40–0.91], $I^2$ = 89%, $p$ < 0.01).

Three studies (*Hu et al., 2022*; *Fischer et al., 2018*; *Eskelinen et al., 2009*) have investigated the effects of tea intake or consumption in people whose BMI is 25 or higher, and the results suggest that tea intake or consumption could reduce the risk of all-cause dementia by 30% (Table 2, RR = 0.70, 95% CI [0.43–1.15], $I^2$ = 90%, $p$ < 0.01). Equally, three studies (*Matsushita et al., 2021*; *Chuang et al., 2019*; *Tomata et al., 2016*) have reported similar results for those with a BMI less than 25 (Table 2, RR = 0.70, 95% CI [0.60–0.80], $I^2$ = 0%, $p$ = 0.63), indicating that the benefits of tea consumption in relatively light and heavy populations.

In three studies (*Matsushita et al., 2021*; *Fischer et al., 2018*; *Eskelinen et al., 2009*) that included over 20% smokers, tea intake can reduce the risk of dementia by a substantial 35% (Table 2, RR = 0.65, 95% CI [0.40–1.06], $I^2$ = 91%, $p$ < 0.01). In four studies (*Hu et al., 2022*; *Chuang et al., 2019*; *Tomata et al., 2016*; *Noguchi-Shinohara et al., 2014*) that included less than 20% smokers, tea intake can reduce the risk of dementia by 34% (Table 2, RR = 0.76, 95% CI [0.64–0.89], $I^2$ = 53%, $p$ = 0.53). These findings suggest that tea intake is more beneficial for reducing the risk of dementia in smokers.

Four studies (*Hu et al., 2022*; *Chuang et al., 2019*; *Fischer et al., 2018*; *Noguchi-Shinohara et al., 2014*) included over 60% tea drinkers who engage in middle or high levels of physical activity. In these studies, tea intake can reduce the risk of dementia by 33% (Table 2, RR = 0.77, 95% CI [0.54–1.11], $I^2$ = 71%, $p$ = 0.02). Conversely, in two studies (*Tomata et al., 2016*; *Eskelinen et al., 2009*) that included less than 60% tea drinkers who engage in middle or high levels of physical activity, tea intake can reduce the risk of dementia by 45%

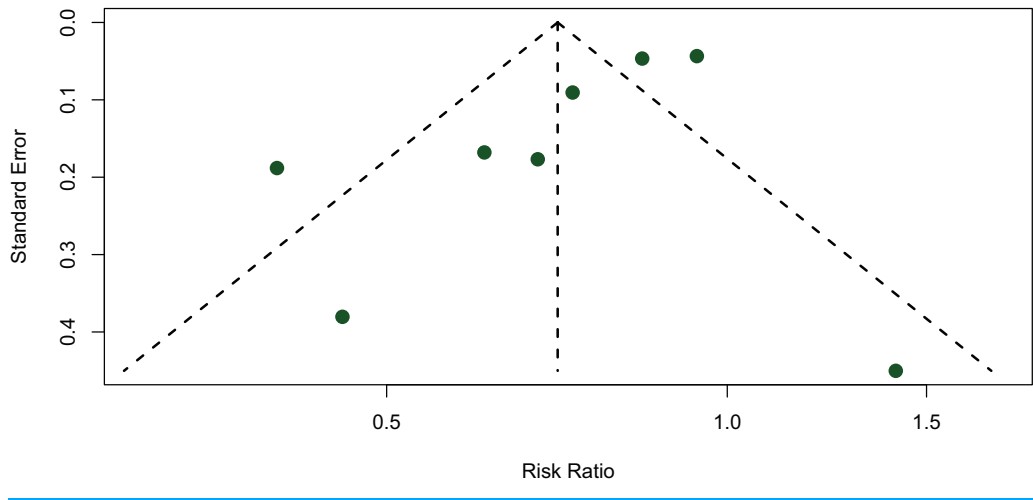

**Figure 4 Funnel plot.**               

(Table 2, RR = 0.55, 95% CI [0.31–0.99], $I^2$ = 88%, $p$ < 0.01). The data suggests that the protective effects of tea may vary to a great degree depending on the level of physical activity.

## Publication bias

The funnel plot was used to assess the publication bias. Although there is some degree of asymmetry observed by the naked eye. However, the $p$ value obtained by the Egger's regression test was 0.0848 ($p$ = 0.0848), indicating there was no significant evidence of publication bias (Fig. 4).

## DISCUSSION

### Main findings

This meta included seven prospective cohort studies involving 410,951 participants, which has provided strong evidence linking tea intake or consumption to a lower risk of all-cause dementia, mainly including AD and VD. The results showed that tea intake or consumption was associated with a remarkable decrease in the risk of all-cause dementia by as much as 29%, 12%, or more than 25%, which indicates that tea intake or consumption may have a preventive effect on dementia. It is consistent with previous studies. A systematic review which enrolled several cross-sectional and longitudinal population-based studies demonstrated a protective effect of coffee, tea, and caffeine against cognitive impairment regression in later life, further supporting our findings (*Panza et al., 2015*).

Our subgroup analysis has revealed fascinating insights into the effects of tea consumption on specific population groups. We found that drinking tea had a more beneficial effect in reducing the risk of dementia among populations with less physical activity, older age, APOE carriers, and smokers. However, interestingly, the data indicated that a higher frequency of tea drinking was associated with a reduced protective effect

against dementia. These findings provide valuable information for clinicians and public health officials seeking to develop preventative strategies against dementia.

## Interpretation of findings

Based on the degree of oxidation and prevalence, tea can be categorized into green tea (non-oxidized), black tea (fully oxidized), white tea (lightly oxidized), and oolong tea (partially oxidized) (*Chupeerach et al., 2021*; *Sentkowska & Pyrzynska, 2022*). Despite their different fermentation degree and production process, all teas contain many bioactive components such as polyphenols, theanine, caffeine, and theaflavins, which have the potential to affect the pathophysiological mechanism of AD and VD (*Chen et al., 2018*). One of the main components of tea polyphenols, EGCG, has been studied for its potential in regulating β-Secretase, γ-Secretase, and amyloid precursor protein (APP) to reduce toxic levels of Aβ (*Payne et al., 2022*; *Pervin et al., 2018*; *Yuksel & Tacal, 2019*; *Mazumder & Choudhury, 2019*) Through hydrogen bond interactions, EGCG can alter the shape of Aβ42 and disrupt Aβ42 protofibril, thereby reducing the formation of Aβ42 plaques in the brain (*Zhan et al., 2020*) Animal experiments have also shown that EGCG can significantly improve cognitive impairment in aged rats (*Wei et al., 2019*). Green tea catechins, particularly EGCG, have been found to have significant neuroprotective effects. EGCG has the ability to boost antioxidant capacity by elevating the mRNA expression level of glutamyl cysteine ligase, thereby resulting in elevated levels of glutathione (*Kim et al., 2009*). Furthermore, research has indicated that EGCG possesses the capability to suppress astrocyte proliferation, impede the expression of glial fibrillary acidic protein, diminish microglial activation, safeguard against neuronal loss (*Rrapo et al., 2009*), and mitigate the production of pro-inflammatory cytokines (*Ren et al., 2018*) EGCG-Zn, a chelated form of EGCG, has been found to improve learning, memory, and antioxidant abilities in VD rats, (*Guo et al., 2017*) while other research has shown that tea polyphenols can improve learning and memory function in VD rats through protecting neurons and regulating the expression of Ach (*Sathya & Devi, 2018*). Additionally, saponin E1, a component of tea, may activate the α-secretase to reduce Aβ concentration and the activity of acetylcholinesterase, which can reduce oxidative stress and regulate signaling pathways and metal chelation (*Khan et al., 2020*). Theanine, a bioactive compound present in tea, demonstrates neuroprotective effects by inhibiting glutamate receptors and regulating extracellular glutamine levels. It also promotes nerve nourishment and facilitates the repair and regeneration of nerve cells (*Xie et al., 2022*). Additionally, L-theanine has the potential to mitigate protein oxidation and lipid damage in the brain by inhibiting ERK/p38 mitogen-activated protein kinase and NF-κB or by elevating glutathione levels. These findings indicate the potential of L-theanine in the prevention and management of Alzheimer's disease (*Kim et al., 2009*; *Williams et al., 2019*). Furthermore, theanine has been found to modulate the effects of the excitotoxin glutamate and influence extracellular glutamine levels, which are regulated by the glutamine transporter expressed on astrocytes (*Kakuda et al., 2008*). Moreover, caffeine and theaflavins exhibit neuroprotective mechanisms, potentially through their ability to counteract the adenosine receptor A2AR and their antioxidant properties. Research has also demonstrated the ability of green tea

catechins and black tea theaflavin to attenuate microglial activation in a model of activated microglial cells induced by Aβ. Notably, EGCG has shown effective blood-brain barrier penetration and significant reduction in the expression of TNFα, IL-1β, IL-6, and iNOS (*Farkhondeh et al., 2020*). Theaflavins have also been shown to suppress production of inflammatory cytokines, prevent dendritic atrophy, and prevent spine loss in the brain (*Ano et al., 2019*). Finally, tea polyphenols can improve learning and memory function in VD rats by protecting neurons and regulating the expression of acetylcholine (*Li et al., 2017*) Green tea polyphenols may act as cholinesterase inhibitors to enhance acetylcholine levels by binding to butyrylcholinesterase (*Ali et al., 2016*).

This article examined the impact of tea intake or consumption on dementia, AD, and VD. We analyzed studies from various countries, including Asian countries (China and Japan), as well as European countries (Britain, Germany, and Finland). However, studies from China and Finland did not differentiate between types of tea, while others from Japan and Germany focused on green and black tea, respectively. Additionally, some studies did not classify the type of tea, making it difficult for us to conduct a subgroup analysis. Our findings suggest that tea intake or consumption may reduce the risk of dementia, AD, and VD. However, two studies showed no association between tea and dementia, possibly due to the small proportion of tea drinkers in these studies (86 black tea drinkers among 490 participants in the Noguchi-Shinohara study and 32.3% tea drinkers in the Fischer study) (*Ren et al., 2018*). It's worth noting that tea-drinking preferences vary greatly between Asian and European/American countries. In general, green tea is more popular in Asian countries, while black tea is preferred in European and American countries. However, there are significant differences in tea consumption even within countries, which may explain some of the differences in study results. Recent research has shown that green tea can improve cognitive function by reducing AD pathology and oxidative stress in middle-aged and older Chinese individuals. Furthermore, a study found that different types of tea, including oolong, white, green, black, dark, and yellow tea, can all prevent the formation of aging-related amyloid protein (*Zhang et al., 2022*). In another study, (*Baranowska-Wójcik, Szwajgier & Winiarska-Mieczan, 2020*) the effect of brewing conditions on the inhibition of AChE activity in tea extracts (black tea, white tea) was evaluated. The study revealed that neither temperature nor brewing time affected the ability of tea to inhibit AChE activity. The anticholinesterase activity was observed in most types of tea analyzed, indicating that tea can be a potential source of this beneficial activity. A cell experiment (*Li et al., 2019*) demonstrated that each tea type (oolong, white, green, black tea) inhibited the formation of Aβ aggregates. Similarly, a mouse animal study (*Wan et al., 2021*) found different types of tea, including black tea, green tea, dark tea, yellow tea, white tea, and oolong tea, could avoid the formation of aging-related amyloid protein, with oolong tea being the most active *in vitro*. Another animal study confirmed that both black and green tea could protect against oxidative stress damage in the hippocampus, with green tea being the better option (*Schimidt et al., 2017*). Previous review investigated the association between green tea intake and dementia, or cognitive impairment, and found that green tea intake might reduce the risk of these conditions (*Kakutani, Watanabe & Murayama, 2019*). Based on these findings, we can conclude that drinking different types
of tea, such as green tea and black tea, can reduce the risk of all-cause dementia, AD, and VD, making it a healthy beverage choice.

The relationship between tea consumption and the risk of dementia is a complex topic that requires careful examination. One important factor to consider is the dose effect, which refers to the amount of tea consumed and its impact on dementia risk. Our study examined seven previous studies on tea and dementia risk, but we found that there was no standard description of tea intake among these studies. Some studies described tea intake in terms of the size of the cup, while others used frequency to describe how often tea was consumed. As a result, it was difficult to conduct a subgroup analysis based on tea intake. However, a large cross-sectional study (Shen et al., 2015) from China showed that consuming 2–4 or more than four cups of tea per day was associated with a reduced risk of cognitive impairment in the elderly. The ideal tea intake was found to be 250 ml per cup. This finding is consistent with the results of our study, which found that the average tea intake among the studies we examined was three cups per day. Additionally, a dose-response meta-analysis (Ran et al., 2021) of 29 multinational prospective studies found that drinking one cup of tea per day reduced the risk of cognitive decline by 6%. Other studies also support the idea that increasing tea intake can reduce the risk of dementia. For example, drinking an extra cup of tea per day reduced the incidence of dementia by 6%, according to Hu et al. (2022). However, Zhang et al. (2022) found that the lowest risk of dementia was at the level of 0.5 to one cup of coffee and four cups of tea per day. Our subgroup results partly reflect this point, that is, although we found that tea intake can reduce the risk of dementia, high frequency of tea drinking may actually reduce the ability to lower the risk of dementia. This result is similar to the findings of three studies by Hu study, Matsushita study, and Eskelinen study, (Hu et al., 2022; Matsushita et al., 2021; Eskelinen et al., 2009) which all showed that moderate tea intake had the best ability to reduce the risk of dementia, while excessive and frequent tea drinking as well as excessive daily tea consumption may actually decrease the ability to lower the risk of dementia. Based on our own findings and previous studies, we recommend a three-cup tea intake to reduce the risk of all-cause dementia, Alzheimer's disease, and vascular dementia. While each of the cohort studies we examined adjusted for confounding factors, the specific factors varied between studies. Therefore, it is impossible to completely eliminate the impact of confounding factors on our study. However, by combining different opinions and studies, we were able to draw a more accurate and reliable conclusion. Our study provides evidence that tea intake or consumption can significantly reduce the risk of dementia, and that the age of tea intake or consumption can also impact this risk. These findings are particularly noteworthy because previous meta-analyses did not identify this meaningful result (Ma et al., 2016). This may be due to the fact that we included more high-quality and larger sample prospective cohort studies, which were better able to control for confounding factors and produce more reliable results.

## Advantages and limitations

This study provides evidence that tea intake or consumption can help prevent all-cause dementia, AD, and VD, regardless of age. This finding is significant because it could

contribute to the development of public measures to prevent dementia. The study has some following strengths: the report adheres to the MOOSE checklist and registered on PROSPERO to ensure transparency and scientific validity. Additionally, including only prospective cohort studies helped reduce interference from recall bias and other confounding factors. However, there are also limitations to this study. Firstly, only cohort studies were included due to a lack of randomized controlled experiments and case-control studies. Secondly, the self-reported measurements of tea intake or consumption did not account for the standardized sizes of specific intake cups, and there was no subsequent assessment of tea consumption over time after the baseline assessment. This limitation neglected some potential confounding factors of drinking frequency and time. Thirdly, the study did not conduct subgroup analysis on the types and intake volume of tea. Different types of tea have different processing procedures, which results in varying bioactive ingredients content. Thus, the lack of specific tea types division in the included studies prevented the analysis of differences between different types in all-cause dementia, AD, VD using tea types as subgroups. As a result, further studies exploring the associations between different types of tea and all-cause dementia, AD, VD are necessary to validate the conclusions.

## CONCLUSIONS

In conclusion, this meta-analysis has found evidence that tea intake or consumption, specifically green tea and black tea, can help reduce the risk of all-cause dementia, Alzheimer's disease and vascular dementia. The study shows that middle-aged and elderly individuals who incorporate tea into their daily routine may have a lower risk of developing dementia. This suggests that making a simple lifestyle change such as drinking tea could potentially help prevent the onset of dementia. Overall, this study highlights the potential benefits of a simple and accessible habit that could have a significant impact on brain health.

### Funding

This work was supported by the National Natural Science Foundation of China (No. 61263033), the International Science and Technology Cooperation Project of Hainan (No. KJHZ2015-4), the Higher School Scientific Research Project of Hainan Province (No. Hnky2015-80), the Research program of Medical and Health Science and Technology Development Plan Project of Shandong Province (No. 202103070653), the Natural Science Foundation of Shandong Province (No. ZR2022MH124), the Youth Science Foundation of Shandong First Medical University (No. 202201-105), the Shandong Medical and Health Technology Development Fund (No. 202103070325), the Shandong Province Traditional Chinese Medicine Science and Technology Project (No. M-2022216) and the Nursery Project of the Affiliated Tai'an City Central Hospital of Qingdao University (No. 2022MPM06). The funders had no role in study design, data collection and analysis, decision to publish, or preparation of the manuscript.

## Grant Disclosures

The following grant information was disclosed by the authors:

National Natural Science Foundation of China: 61263033.
International Science and Technology Cooperation Project of Hainan: KJHZ2015-4.
Higher School Scientific Research Project of Hainan Province: Hnky2015-80.
Research program of Medical and Health Science and Technology Development Plan Project of Shandong province: 202103070653.
Natural Science Foundation of Shandong Province: ZR2022MH124.
Youth Science Foundation of Shandong First Medical University: 202201-105.
Shandong Medical and Health Technology Development Fund: 202103070325.
Shandong Province Traditional Chinese Medicine Science and Technology Project: M-2022216.
Nursery Project of the Affiliated Tai'an City Central Hospital of Qingdao University: 2022MPM06.

## Competing Interests

Yuzhen Xu is an Academic Editor for PeerJ.

## Author Contributions

- Ning Jiang performed the experiments, analyzed the data, prepared figures and/or tables, and approved the final draft.
- Jinlong Ma performed the experiments, analyzed the data, prepared figures and/or tables, and approved the final draft.
- Qian Wang analyzed the data, authored or reviewed drafts of the article, and approved the final draft.
- Yuzhen Xu conceived and designed the experiments, performed the experiments, prepared figures and/or tables, authored or reviewed drafts of the article, and approved the final draft.
- Baojian Wei conceived and designed the experiments, performed the experiments, prepared figures and/or tables, authored or reviewed drafts of the article, and approved the final draft.

## Data Availability

The raw measurements are available in the Supplemental Files.

## Supplemental Information

Supplemental information for this article can be found online at http://dx.doi.org/10.7717/peerj.15688#supplemental-information.

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
