# Peer review of "Tea intake or consumption and the risk of dementia: a meta-analysis of prospective cohort studies"

_PeerJ, doi:10.7717/peerj.15688_

## Round 0.1 · original submission · Major Revisions

Please revise the manuscript as the reviewers suggested.

·

Basic reporting

Thank you for the conduct of this review. This topic is very important and further global attention is needed towards prevention dementia, as it has detrimental consequences for individuals and society.

Please check the language carefully, there many sentences significant incomplete. And don’t have the period in different sentence.
-1. The first issue is about the topic chosen for the systematic review. Although very important, I believe the relationship between tea and dementia are well-established. As the authors have mentioned in the Introduction, tea has the potential influence on preventing dementia and some prospective cohort studies have proven it. So, what’s the aim to do the meta-analysis.

Experimental design

-2. Method: Why only include dementia and AD in your medical subject terms. Actually, dementia includes many specific types, such as VD, FTD or alcohol related dementia. I don’t think you can find all papers about dementia only using AD and dementia. Thus, I recommended you can include more comprehensive medical subjects terms such as, vascular dementia, frontotemporal dementia or other types of dementia.
-3. Method: You eligible participants included VD, but this term don’t included in your search strategy.
-4. Method: Lang Yanmei and Meng Li extracted the data, why they don’t be listed as co-authors?
-5 Method: The platforms for the databases should be included, e.g., Web of Science, Medline. Cochrane database is for RCT-meta. Please use the suitable databases.
- 6 Method: Ideally, you should the number studies from each databases. Such as, xx from pubmed, xx from web of science.

Validity of the findings

-7 Result: The trials in these studies lasted between 1.5 and 21 years, with an average of 9 years. What does trails mean?
-8 Result: In addition, in data analysis, almost all studies adjusted for age, gender, education, disease history and other confounding factors. Please use number instead of almost all studies.
-9 Result: can you add reference in the table1 . it’s difficult to find the origin papers.
-10 Method and result: I read the inclusion papers in your study. These studies reported consumption of tea compared with no-drinkers and categorical of tea consumption. So what’s data you extracted from the paper?

Additional comments

-11 Discussion: systematic evaluation showing several cross-sectional and longitudinal population-based studies demonstrating a protective effect of coffee, tea, and caffeine use
against cognitive impairment regression in later life, further supporting our findings. This sentence is not complete.
-12 Discussion: In the 4.2 section, the discussion overkills the results. The study is a meta-analyses from prospective studies. But you introduce many biological mechanisms.
-13 Discussion: In the 4.2 section, please discuss around your findings.

Reviewer 2 ·

Basic reporting

1.Background
The introduction and background of this meta-analysis are well done, which clearly introduces the research status of dementia: there is no effective drug for treating dementia, so it also highlights the importance of this research, that is, the importance of prevention, prevention and management of dementia.
2.Methods
2.1 There is a conflict in which cognitive impairment is used in literature search, but it is not mentioned in the inclusion part.
2.2 Data analysis, this article mentioned that it is impossible to conduct subgroup analysis on different kinds of tea and dosage, so how to control the influence of these factors on the results
2.3Inclusion and exclusion criteria: cognitive impairment is included in search terms, why cognitive impairment is not included in the research, and how to select articles from the same database.
3.Discussion
The discussion of this study is very detailed about preventing or reducing the risk of all-cause dementia, AD and vascular dementia by tea, and it also clearly introduces the importance of this study for dementia prevention: Drinking tea may become an important dietary measure to prevent dementia.
4.Conclusion:
Tea intake over sixty-five years old or under sixty-five years old can reduce the risk of all-cause dementia, Alzheimer's disease and vascular dementia. This conclusion seems to be too wide, and the age range of the population included in the cohort study can not be determined, nor can it be defined by the mean and standard deviation. It is suggested to use appropriate wording.

Experimental design

.

Validity of the findings

.

Reviewer 3 ·

Basic reporting

The meta-analysis by Jiang and colleagues compiled studies on whether tea consumption reduces the risk of dementia and other neurodegenerative diseases. Dementia, loss of memory, and thinking ability serve to interfere with daily life. Alzheimer’s is the most common cause of dementia. Both genetic and/or environmental factors play a major role as a causative.
Basic reporting
The manuscript was clear, and professional English language was used throughout the text. The introduction was appropriate, and all the studies are cited accordingly.
Experimental design
This meta-analysis was based on nine studies out of 1545 records between 2009 to 2022 with a total population of 1,128,302 worldwide. The authors then compiled the data systematically to discuss the effect of tea consumption in reducing dementia.
Validity of the findings
There are a lot of studies on tea consumption and calculating the relative risk of dementia and neurogenerative disease. The authors did not discuss any novel findings in the present meta-analysis. Overall conclusion is not convincing.
Specific comments
1. In the methodology, the authors have excluded the studies which did not show any interesting outcomes. The authors can discuss the negative effect of tea consumption on dementia and the population studied. They may have some other factors inhibiting these mechanisms.
2. In line 183, the authors mention the trials lasted between 1.5 and 21 years, but in Table 1 follow-up time was mentioned as ranging from 4.9 to 21 years. Comment on this statement.
3. The major finding of this meta-analysis is the age of tea intake (lines 326-327). Tea consumption from middle age to old age might reduce the risk of dementia. The same statement was mentioned in PMID: 35474192. But there could be a possibility that the study participants have taken the tea from their adulthood. Does this have any long-term effect in preventing the risk of dementia among them? Does any of the studies report the duration of tea consumption by the participants?
4. The total number of participants from these nine reports was 1,128,302. But three studies (Hu et al., Schaefer et al., Zhang et al.,) have utilized the same study participants from the UK biobank collected between 2006 to 2010. The strategy and age group selected slightly differ among the three. The authors can combine the three studies together and consider them as one report.
5. Few studies (Hu et al., Fischer et al., Shinohar et al.,) from different populations have reported the carrier status of the APOE4 gene. APOE4 increases the risk of Alzheimer's disease, and it was reported to be associated with the early stage of disease onset in a particular population. The authors can discuss in this aspect whether there is any relative significance in reducing the risk of dementia or Alzheimer’s among the APOE4 carrier participants after tea consumption.
6. Similarly, there are other factors (BMI, smoking, physical activity, diet) that interfere with the components present in tea in reducing the risk of neurodegenerative diseases.
7. In lines 280-282, the authors mentioned “However, two studies showed no association between tea and dementia, possibly due to the small sample size of participants in these studies (82 participants in the Noguchi-Shinhora study and 45 participants in the Eskelinen study)” but the sample size was mentioned as 723 and 1409 respectively in Table 1. Comment on this.
8. The interpretation of the finding was brief, and the authors can explain in detail the mechanism of how the tea components regulate the pathophysiological mechanism of AD and VD.
9. Lines 189-194 are repeated twice in the respective paragraph.
10. Is there any placebo controlled clinical trial in human subjects that have utilized active component present in tea? Reviewer understands that it takes time to develop clinical symptoms but a short duration may be 6 months supplementation study should help delileate any effect of tea active component on ADAM, PACE and Ab42/40 status in subjects.
11. Line 222: publication bias: p value of 0.054 should not be considered as no evidence of publication bias. It is very close to 0.05 significant.

Experimental design

.

Validity of the findings

.

---

## Round 0.2 · accepted · Accept

This manuscript can be accepted now.